# Impact of Formate Supplementation on Body Weight and Plasma Amino Acids

**DOI:** 10.3390/nu12082181

**Published:** 2020-07-22

**Authors:** Sandeep Dhayade, Matthias Pietzke, Robert Wiesheu, Jacqueline Tait-Mulder, Dimitris Athineos, David Sumpton, Seth Coffelt, Karen Blyth, Alexei Vazquez

**Affiliations:** 1Cancer Research UK Beatson Institute, Glasgow G61 1BD, UK; sandeepdhayade@gmail.com (S.D.); Matthias.Pietzke@mdc-berlin.de (M.P.); r.wiesheu.1@research.gla.ac.uk (R.W.); J.Tait-Mulder@beatson.gla.ac.uk (J.T.-M.); d.athineos@beatson.gla.ac.uk (D.A.); d.sumpton@beatson.gla.ac.uk (D.S.); Seth.Coffelt@glasgow.ac.uk (S.C.); Karen.Blyth@glasgow.ac.uk (K.B.); 2Institute of Cancer Sciences, University of Glasgow, Glasgow G61 1QH, UK

**Keywords:** formate, immune system, body weight, microbiome, metabolomics

## Abstract

Current nutritional recommendations are focused on energy, fat, carbohydrate, protein and vitamins. Less attention has been paid to the nutritional demand of one-carbon units for nucleotide and methionine synthesis. Here, we investigated the impact of sodium formate supplementation as a nutritional intervention to increase the dietary intake of one-carbon units. A cohort of six female and six male mice received 125 mM of sodium formate in the drinking water for three months. A control group of another six female and six male mice was also followed up for the same period of time. Tail vein blood samples were collected once a month and profiled with a haematology analyser. At the end of the study, blood and tissues were collected for metabolomics analysis and immune cell profiling. Formate supplementation had no significant physiological effect on male mice, except for a small decrease in body weight. Formate supplementation had no significant effect on the immune cell counts during the intervention or at the end of the study in either gender. In female mice, however, the body weight and spleen wet weight were significantly increased by formate supplementation, while the blood plasma levels of amino acids were decreased. Formate supplementation also increased the frequency of bifidobacteria, a probiotic bacterium, in the stools of female mice. We conclude that formate supplementation induces physiological changes in a gender-specific manner.

## 1. Introduction

Folic acid, vitamin B_9_, is an essential cofactor in one-carbon metabolism, a network of biochemical reactions mediating the synthesis of methionine and nucleotides [1]. Our cells are unable to produce folic acid, and therefore, we satisfy our daily demand for folic acid through our diet. The nutritional requirements for folic acid have been extensively addressed in the scientific literature [2], including the interaction between folic acid and vitamin B_12_. Methionine synthase requires vitamin B_12_ as a co-factor to transfer one-carbon units from the folate pool to the methyl pool [1]. Deficiencies in either folic acid or vitamin B_12_ lead to a spectrum of similar pathologies [3].

New studies have highlighted the urgent need to investigate the nutritional requirements of the one-carbon units per se [4,5]. One-carbon units can be obtained from the amino acid serine, glycine and methionine; from methyl containing molecules such as choline, methanol, methylamine and creatine; and from the quintessential one-carbon precursor, formate [4,5]. The diversity of one-carbon nutritional sources may give the impression that there is no need to specify the nutritional requirements of one-carbon units. However, our observations of low serum formate levels in cancer patients and obese individuals compared to healthy controls provide a rationale to investigate the distinctive role of formate in health and disease more in depth [6].

Our daily demand of one-carbon units can be estimated from the uric acid content in urine. Uric acid is the product of purine catabolism and it contains two one-carbon units. Based on the daily urine excretion of uric acid, we need approximately 1 g of formate per day [7]. This demand may be satisfied from the consumption of formate precursors contained in animal meats, dietary fibre and coffee [5,7].

In the present study, we investigated the impact of dietary formate intake on mammalian physiology. To this end, we conducted sodium formate supplementation experiments in female and male mice. Our data suggest that sodium formate supplementation has an anabolic effect in female mice, with a significant increase in body weight and a significant reduction of plasma amino acid levels. Sodium formate furthermore increased the frequency of bifidobacteria, a probiotic bacterium, in the gut of female mice. In the male cohort, there was a striking variability from mouse to mouse in their response to sodium formate supplementation. Additionally, we profiled the immune system of female and male mice and observed no significant impact of sodium formate supplementation.

## 2. Experimental Methods

### 2.1. Ethical Statement

In vivo experiments were carried out in dedicated barriered facilities proactive in environmental enrichment under the EU Directive 2010 and Animal (Scientific Procedures) Act (HO licence number: 70/8468) with ethical review approval (University of Glasgow). Animals were cared for by trained and licensed individuals and humanely sacrificed using Schedule 1 methods.

### 2.2. Study Design

Twelve female and twelve male mice were followed up for 127 days, approximately 4 months, from age 56 days to age 183 days. Six female and six male mice received water with 125 mM of sodium formate (MilliporeSigma, USA) and the remaining six female and male mice drank water alone. Mice were randomized in individual groups (different gender and treatment arms).

### 2.3. Experimental Procedures

We first estimated the daily demand of formate for mice. In mammals, formate is incorporated into purines and thymidylate. Purines are metabolised to uric acid, and in mice, uric acid is further converted to allantoin, which is then secreted in the urine [8]. Based on this evidence, we assumed the urinary excretion of allantoin as the major sink of one-carbon units in mice. Under this assumption, the reference daily intake of formate for mice is *r* = 2 *C_A_U*/*M_A_*, where *C_A_* is the concentration of allantoin in mouse urine, *U* is the typical mouse urine volume per day and *M_A_* is the molar mass of allantoin. The concentration of allantoin in the mouse urine is about 3.5 mg/mL [8]. We assumed a urine output of 1 mL/day [9]. The molecular mass of allantoin is 158 g/mol. Based on these numbers, we calculated a demand of 40 mmol formate/day.

If we were to distribute the estimated 40 mmol formate in the typical daily intake of water for our C57BL/6 mice (30 mL), we would obtain a target concentration of 1 M. A pilot study was carried out, supplementing sodium formate at concentrations of 125 mM, 250 mM and 500 mM in the drinking water. Mice receiving the 500 mM dose lost weight, and therefore, we stopped this arm of the pilot study. Mice receiving the 125 mM and 250 mM doses did not manifest any appreciable change in body weight. Based on this evidence, we decided to use the 125 mM dose of sodium formate in our study, which represents about 10% of the daily formate demand for mice.

### 2.4. Experimental Animals

We used mice as a model to investigate the impact of long-term formate supplementation on mammalian physiology. This choice was motivated by a number of factors. First, and most important, murine formate metabolism is representative of mammalian formate metabolism and has been proven to provide insights into the formate metabolism of mammals, including humans. Second, the C57BL/6 mice used in this study had a homogeneous genetic background and they were kept under the same conditions and diet. With this choice, we thus avoided several confounding factors that could have occurred if the study was conducted in the human population. Finally, we aimed to investigate changes in the immune system at different levels, from the thymus and spleen compartments to the blood. This investigation is not feasible in humans.

Wild-type female and male C57BL/6J mice (eight weeks of age) were purchased from Charles River.

### 2.5. Housing and Husbandry

Animals were housed in a purposely built facility proactive in environmental enrichment in individually ventilated cages (Tecniplast, Buguggiate, Italy) housing 3–5 animals. Animals were cared for by highly skilled technicians. For this study, animals were purchased from a recognised breeder Charles River, UK. We use a 12-h light/dark cycle (7 a.m.–7 p.m.). All animals underwent a daily welfare assessment in accordance of the Standard Conditions of the Establishment Licence issued by the UK government. In addition, the researcher carried out experimental health checks (2–3 times weekly during the study).

### 2.6. Sample Size

In a pilot experiment investigating five female mice drinking water supplemented with 125 mM of sodium formate and three female control mice drinking water alone, we observed blood lymphocyte counts of 10 millions/mL and 6 millions/mL for mice supplemented with formate and controls, respectively, with a standard deviation of 2 millions/mL. Based on these numbers, a statistical power of 80% and two-sided significant level of 0.05, the estimated sample size is *n* = 4 for each group (calculated with the online tool at https://www.stat.ubc.ca/~rollin/stats/ssize/n2.html). Here, we used six mice per group.

### 2.7. Allocating Animals to Experimental Groups

Age-matched mice were randomized in individual groups (different gender and treatment arms). All mice were of the same age.

### 2.8. Experimental Outcomes

Tail vein blood samples were taken once every month. Samples were analysed on an IDEXX ProCyte Dx Haematology Analyser.

At the experimental endpoint, mice were sacrificed humanely by standard Schedule 1 method. Blood was immediately taken by cardiac puncture, transferred into Eppendorf tubes and centrifuged at 4 °C for 10 min at 13 kG. The supernatant was transferred into new Eppendorf tubes and flash frozen in liquid nitrogen. The spleen was weighed. Bone marrow was harvested by flushing one femur per specimen with ice-cold phosphate-buffered saline (PBS) and kept on ice until further procession. Tissues were split in half and then harvested in Eppendorf tubes and flash frozen in liquid nitrogen for metabolomics. For further analysis by multiparameter flow cytometry, samples were kept in PBS on ice. Flash frozen tissue samples were blinded with random IDs and processed by a different person for metabolite extraction and analysis. After final data analysis IDs were uncovered.

Plasma formate was measured as a benzyl-derivative. An Agilent 7890B GC was used for the measurements, equipped with a Phenomenex ZB-1701 column (30 m × 0.25 mm × 0.25 µm) coupled to an Agilent 7000 triple-quad-MS. The temperature of the inlet was 280 °C, the interface temperature was 230 °C, and the quadrupole temperature was 200 °C and the EI voltage was set to 60 eV. Next, 2 µL of the sample was injected and transferred to the column in split mode (1:25) with a constant gas flow through the column of 1 mL/min. The oven temperature started at 60 °C, was held for 0.5 min, followed by a ramp of 38 °C/min to 230 °C, which was held for another minute. The total run time was 6 min and the retention time of benzyl-formate was 3.8 min. The mass spectrometer was operated in selected ion monitoring (SIM) mode between 3.0 min and 4.3 min with SIM masses of 136, 137 and 138 for M0, M + 1 and M + 2 (internal standard) formate, respectively. Recorded data were processed using the MassHunter Software (Agilent, Santa Clara, CA, USA). Integrated peak areas were extracted and used for further quantifications using in-house scripts, including the subtraction of natural isotope abundances, as described by the authors of [10] for formate or in the Appendix A for formaldehyde.

Plasma metabolite extraction for liquid chromatography-mass spectrometry (LC-MS) analysis was performed as previously described [11]. LC-MS analysis was performed as described previously using hydrophilic interaction liquid chromatography (pHILIC) and a Q-Exactive mass spectrometer (Thermo Fisher Scientific, Waltham, MA, USA). Raw data analysis was performed using TraceFinder (Thermo Fisher Scientific) software.

For the flow cytometry immune cell analysis, single-cell suspensions of all tissues were prepared by mashing samples through a 70-µm cell strainer. Erythrocytes were lysed with ammonium chloride lysis buffer (10× RBC Lysis Buffer, eBioscience, San Diego, CA, USA). For T cell activation and cytokine staining, cells were stimulated with Cell Activation Cocktail with Brefeldin A (Biolegend, San Diego, CA, USA, 1:500) in IMDM supplemented with 8% FCS, 0.5% 2-Mercaptoethanol and penicillin-streptomycin solution (MilliporeSigma, Burlington, MA, USA) for 3 h at 37 °C. Single-cell suspensions were further incubated with TruStain FcX (Biolegend) for 20 min at 4 °C to block Fc receptors, followed by incubation with fluorochrome-conjugated antibodies for 30 min at 4 °C in the dark in brilliant stain buffer (BD Biosciences, Franklin Lakes, NJ, USA). Prior to fixation and permeabilization for intracellular staining with Cytofix/Cytoperm (BD Bioscience), cells were stained with Zombie NIR Fixable Viability kit (Biolegend) (1:400, in PBS) for 20 min at 4 °C in the dark for dead cell exclusion. Fluorochrome-conjugated antibodies for intracellular epitopes were diluted in permeabilization buffer and incubated for 30 min at 4 °C in the dark. Samples were acquired using BD LSR II flow cytometer and Diva software. Data analysis was performed by using FlowJo (FlowJo, Ashland, OR, USA) version 9.9.6 and fluorescence minus one (FMO) controls to facilitate gating. Gating strategies for respective immune cell populations can be seen in Appendix A. For calculations of absolute numbers, 123count eBeads (eBioscience) were used according to manufacturer’s instructions.

Antibodies used for spleen, thymus and blood samples: CD19 (eBioscience, clone 1D3, APC-eFluor780, 1:400), EpCAM (eBioscience, clone G8.8, APC-eFluor780, 1:100), CD3 (Biolegend, clone 17A2, BV650, 1:100), CD4 (Biolegend, cloneRM4-5, BV605, 1:100), CD8 (BD Bioscience, clone 53-6.7, BUV395, 1:100), TCRδ (eBioscience, clone GL3, FITC, 1:200), CD11b (eBioscience, clone M1/70, BV785, 1:800), CD27 (Biolegend, clone LG.3A10, PE-Dazzle594, 1:400), CD44 (Biolegend, clone IM7, PerCP-Cy5.5, 1:100), CD69 (Biolegend, clone H1.2F3, BV510, 1:50), NK1.1 (Biolegend, clone PK136, BV421, 1:50), IFNγ (eBioscience, clone XMG1.2, PE-Cy7, 1:200), IL-17A (eBioscience, clone eBio17B7, PE, 1:100), Granzyme B (Biolegend, clone GB11, AF647, 1:50).

Antibodies used for bone marrow analysis: CD4 (Invitrogen, clone GK1.5, APC-eFluor780, 1:100), CD8 (Invitrogen, clone 53-6.7, APC-eFluor780, 1:100), CD11b (eBioscience, clone M1/70, APC-eFluor780, 1:800), CD19 (eBioscience, clone 1D3, APC-eFluor780, 1:400), Ter-119 (eBioscience, clone TER-119, APC-eFluor780, 1:50), CD11c (Biolegend, clone N418, PerCP/Cy5.5, 1:800), CD3ε (eBioscience, clone 145-2C11, FITC, 1:100), Ly-6A/E (Sca-1) (Biolegend, clone D7, BV605, 1:100), NK1.1 (Biolegend, clone PK136, BV421, 1:50), CD117 (c-Kit) (Biolegend, clone 2B8, APC, 1:50), CD127 (IL-7Rα) (Biolegend, clone A7R34, PE, 1:100).

For the microbiome profiling, stool samples were collected at endpoint and stored in Eppendorf tubes containing DNA/RNA shield (Zymo Research). Samples were sent in dry ice to CosmoID for 16S sequencing and inference of microbiome abundance.

### 2.9. Statistical Methods

Reported statistical significances were determined using a *t*-test with two tails and unequal variance, also known as a Welch’s test, using the Octave function welch_test. To assess normality, we used the Shapiro–Wilk test using the shapiro.test function in R. A test was dimmed significant if the data for both the control and the formate-treated samples passed a Shapiro-Wilk test of normality (>0.05) and the Welch’s test for equal means between case and controls failed (<0.05).

## 3. Results

### 3.1. Drinking Rate

From daily weight measurements of the drinking water bottles, we estimated the drinking rate of the female and male cohorts supplemented with formate in the drinking water. We observed no significant differences in the drinking rate per body weight between the female and male mice (Figure 1A). Based on the drinking rate and the concentration of sodium formate in the drinking water, we attained a sodium formate supplementation dose of 9.3 g/day/kg of body weight. This value is close to the mouse LD50 for a single oral dose of sodium formate dissolved in water at 4.7 g/kg [12]. However, the sodium formate dose in our study would have been distributed throughout the 24 h cycle with ad libitum administration, most likely falling below the LD50 for a single dose. No adverse effects were observed during the course of our experiments.

### 3.2. Plasma Formate

Sodium formate supplementation had no significant effect on the plasma formate concentration of female mice (Figure 1B). Male mice drinking water alone had significantly lower plasma formate levels than aged matched female mice and the plasma formate concentration was increased by sodium formate supplementation (Figure 1B). The latter was not significant due to the high variability across the male mice supplemented with sodium formate.

### 3.3. Body Weight

Sodium formate supplementation resulted in a significant increase in the body weight of female mice when compared to female mice drinking water alone (Figure 1C). Because of the follow-up immunological characterization, we also measured the wet weight of spleens at the end of the study. Sodium formate supplementation resulted in a significant increase in the wet spleen weight of female mice when compared to control female mice (Figure 1D). In contrast to female mice, we observed a small but significant decrease in the body weight of male mice (Figure 1D) and no significant changes in the weight of their spleens at the end of the study (Figure 1D).

A hypothesis, for the observed increase in the spleen weight of female mice supplemented with sodium formate relative to controls, is that the increase in body weight was accompanied by an increase in organ weight. To test this hypothesis, we calculated the spleen weights as percent of body weight. There were no significant differences between the percental spleen weight of female mice supplemented with sodium formate relative to controls (Figure 1E), in support of the hypothesis.

### 3.4. Plasma Metabolites

To determine which metabolic pathways might be associated with the observed body weight changes, we analysed the plasma samples using liquid chromatography and high-resolution mass spectrometry. We performed a targeted quantification of selected metabolites. The most striking observation was a significant reduction of all proteogenic amino acids in the plasma of female mice subject to sodium formate supplementation (Figure 2A), while no significant changes were detected in the male cohorts (Figure 2B). This is illustrated in Figure 2C,D for the amino acids threonine and methionine. Sodium formate supplementation also increased the levels of citrate in the plasma of female mice (Figure 2E). The increase in citrate suggests an increase in TCA cycle metabolites.

Additionally, we observed a significant reduction of Timonacic in the plasma of female mice (Figure 2F). Timonacic is an adduct formed from the reaction of formaldehyde with cysteine [13] that has been recently identified in mammals [14]. A reduction in the plasma levels of Timonacic could be due to a reduction on the plasma levels of cysteine. Another possibility is that sodium formate induces a decrease in plasma or tissue formaldehyde and as a consequence a decrease in the levels of Timonacic, the adduct with cysteine. Since formaldehyde is a known carcinogen [15], these data suggest a protective effect of sodium formate supplementation with respect to cancer risk.

### 3.5. Immune System

A recent study reported an age-dependent decline of the availability of one-carbon units in T cells [16]. Based on this evidence, we hypothesised that sodium formate supplementation may have a beneficial effect on the immune system. To obtain a longitudinal characterization of the immune system, we collected tail vein blood samples once per month from each mouse and investigated whole blood leukocyte counts for both female and male cohort animals. Sodium formate supplementation had no significant effect on the counts of any immune cell subpopulation, as shown in Figure 3A–D for lymphocyte and neutrophil levels.

We performed a more extensive characterisation of the immune system at endpoint, including blood samples and tissues from the bone morrow (BM), the thymus and the spleen. Except for a decrease of common lymphoid progenitor cells (CLPs) in the bone marrow of female mice supplemented with sodium formate (Figure 3E), sodium formate supplementation had no other significant effect on the amount of major lymphocyte populations (CD3^+^ and NK cells) in the bone marrow, thymocytes, spleen lymphocytes or blood lymphocytes (Figure 3E–K). We furthermore investigated the absolute numbers of distinct T cell subpopulations, including CD4^+^, CD8^+^ and γδ T cells, as well as the numbers of NK cell subsets based on their maturity. This analysis revealed similar numbers of each cell subset between the control group and formate-treated mice (data not shown). Of note, NK cell numbers were significantly higher in female mice compared to male mice in the thymus and spleen, but these differences remained unaffected by formate supplementation (Figure 3I–K). In addition to absolute number quantification, we scrutinised the activation status of T cell and NK cell subsets based on the expression of CD44 and CD69 alongside their cytokine production (IL-17A, IFNγ and Granzyme B). We detected the same levels of activation marker expression and cytokines for all populations between control groups and sodium formate-supplemented groups.

### 3.6. Microbiome

Formate is a central molecule in the metabolism of the gut microbiome. Formate is a by-product of the anaerobic fermentation by gut bacteria [17], and it can be a substrate for oxidative phosphorylation by pathogenic bacteria like *E. coli* [18]. Based on this evidence, we conducted a characterisation of the gut microbiome.

Stool samples were collected at the endpoint and sent to CosmoID for microbiome profiling. Sodium formate supplementation caused significant changes in the phylogenetic tree of the females microbiome but not of the male’s microbiome (Figure 4A,B). At the family level, sodium formate supplementation increased the frequency of *Bifidobacteriaceae* in the stools of female mice but not of male mice (Figure 4C). Since bifidobacteria are probiotic bacteria with a positive impact on health, these data indicate that sodium formate supplementation has a positive effect on the microbiome of female mice.

## 4. Discussion

We present evidence that sodium formate supplementation induces physiological changes in a gender specific manner. In the case of females, sodium formate supplementation induced an increase in body weight, a concomitant reduction of plasma amino acids and an increase of the frequency of bifidobacteria in the gut microbiome. In contrast, we did not detect any significant effect of sodium formate supplementation in male mice, apart from a small decrease in body weight.

Based on work with cultured cancer cells, formate is an inducer of anabolism [19]. This potentially explains the increase in body weight in female mice supplemented with formate. Consequently, the increase in anabolism shifts the amino acids balance toward an increase in demand, which is in line with our observation of a decrease in plasma amino acids.

The increase in the frequency of bifidobacteria in the gut of female mice was most likely an independent event. In bifidobacteria, there is a dynamic equilibrium between oxalate + formyl-CoA and formate + oxalyl-CoA. In theory, an increased concentration of formate in the gut, as expected by formate supplementation, should reduce the metabolism of oxalate. However, whether this would represent a growth advantage to bifidobacteria remains unknown and warrants further investigation.

To date, gender-specific factors modulating the impact of sodium formate supplementation have not been identified. One possibility is that female mice have a higher one-carbon units demand than male mice and that the number of one-carbon units they receive through the standard chow diet is below their demand. In mammals, circulating formate reaches highest levels during pregnancy [20,21]. The female mice in the cohorts investigated in this study were not pregnant, but they were in their reproductive age. It has been observed that molecules modulating oestrogen metabolism induce the expression of serine synthesis genes in the liver of male mice [22]. The underlying mechanism of this association is not clear yet. However, serine is a source of formate, raising the potential link of female hormones and formate metabolism.

Based on our data for female mice, formate supplementation lowers plasma amino acids. In turn, there is evidence that a low-protein diet induces the expression of genes from the serine synthesis and mitochondrial serine metabolism to glycine and formate in rat livers [22]. This increase at the gene expression level is reflected in an increase of plasma levels of serine and glycine [23], while formate was not quantified. Based on this evidence, we speculate that amino acids deprivation and formate supplementation may trigger similar physiological effects. A low-protein diet has health benefits similar to calorie restriction based on experiments conducted in mice [24]. It is tentative to speculate that formate supplementation could achieve the same positive effect. Further work is required to test this hypothesis.

One aim of this study was to investigate the impact of sodium formate supplementation on the immune system. Surprisingly, we observed no significant effect. Lymphocytes isolated from aged mice exhibit an impaired activation due to a decline in their capacity to catabolize serine to formate in vitro, which can be rescued by formate supplementation [16]. We should bear in mind that our analysis was restricted to the immune system of antigen-naïve mice. At present, we cannot exclude that sodium formate supplementation may have an impact on the immune response in the context of infection or nutritional deficiency.

The extrapolation of these observations to humans requires some discussion. There is evidence of ocular toxicity and death associated with high levels of formate in the context of methanol intoxication [25,26,27]. In contrast, a few grams of formate per day are deemed safe [7]. A study supplementing healthy women with 1300 mg of calcium formate three times a day for 14 days reported no evidence for ocular toxicity [28]. In our study, we set the supplementation to 10% the daily one-carbon demand of mice. A similar calculation can be performed in humans based on the concentration of uric acid in the urine, obtaining a daily demand of 1 g/day or 20 mmol/day [7]. The formate supplementation dose of 10% for the human daily demand translates to 2 mmol of formate per day, or 136 milligrams of sodium formate per day. This dose is considered safe based on the toxicity study cited above. Therefore, our study provides evidence that there is no objection for the recapitulation of a formate supplementation study in humans.

## 5. Conclusions

We conclude that sodium formate supplementation induces physiological changes in a gender-specific manner. Further work is required to determine whether sodium formate supplementation could have similar effects in humans.

## Figures and Tables

**Figure 1 nutrients-12-02181-f001:**
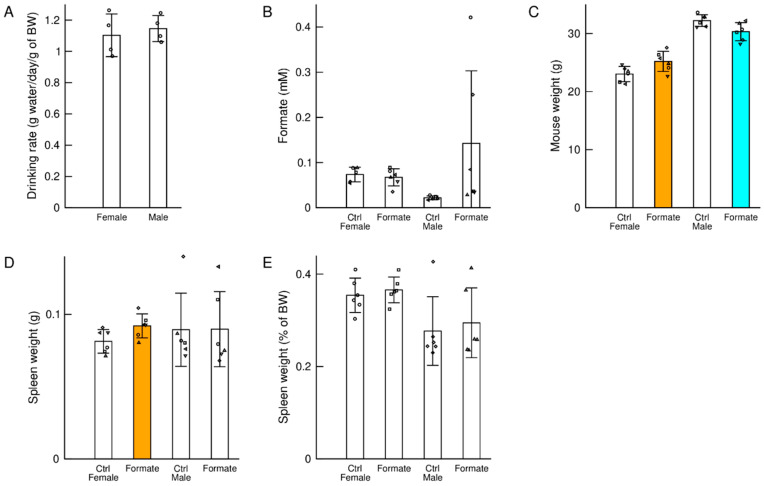
Major characteristics of the female and male cohorts drinking regular water (Ctrl) or water supplemented with 125 mM of sodium formate for 127 days, approximately 4 months. (**A**) Water drinking rate in the female and male cohorts of mice supplemented with sodium formate. (**B**) Plasma formate at the end of the study. (**C**) Body weight at the end of the study. (**D**,**E**) Spleen wet weight at the end of the study, in grams or as % of body weight (BW). Statistics: Each symbol shape represents an individual mouse, the bars heights represent the mean and the error bars the standard deviation. Coloured bars highlight a significant change (*p* < 0.05) in the cohort with sodium formate supplementation relative to control of the same gender. Orange indicates a significant increase and cyan a significant decrease.

**Figure 2 nutrients-12-02181-f002:**
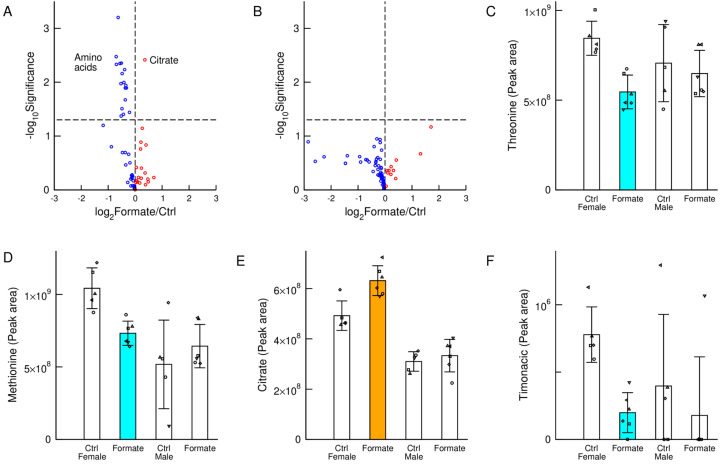
Metabolic changes. (**A**,**B**) Volcano plots of metabolic differences between plasma samples of mice supplemented with 125 mM of sodium formate and controls, for female (**A**) and male mice (**B**). The horizontal axis reports the log-ratio and the vertical axis the statistical significance (two tails and unequal variance *t*-test) in a logarithmic scale. (**C**–**F**) Levels of the indicated metabolites across the different cohorts. Statistics: Each symbol represents an individual mouse, the bar heights represent the mean and the error bars the standard deviation. Coloured bars highlight a significant change (*p* < 0.05) in the cohort with sodium formate supplementation relative to control of the same gender. Orange indicates a significant increase and cyan a significant decrease.

**Figure 3 nutrients-12-02181-f003:**
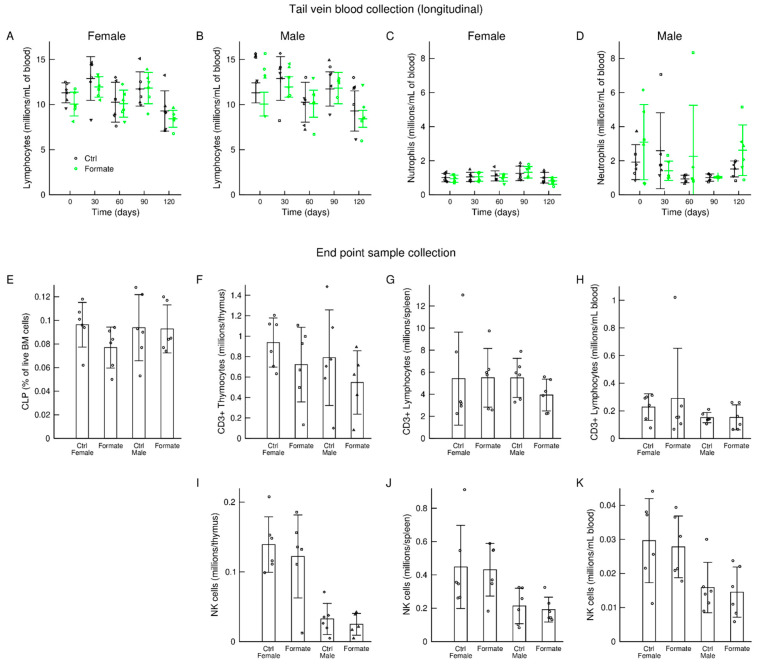
Immune system characterization. (**A**–**D**) Longitudinal counts of immune system subpopulations in blood samples from the tail vein. (**E**–**K**) Immune cell counts from samples at the end of the study. Statistics: Each symbol represents an individual mouse, the bars heights represent the mean and the error bars the standard deviation.

**Figure 4 nutrients-12-02181-f004:**
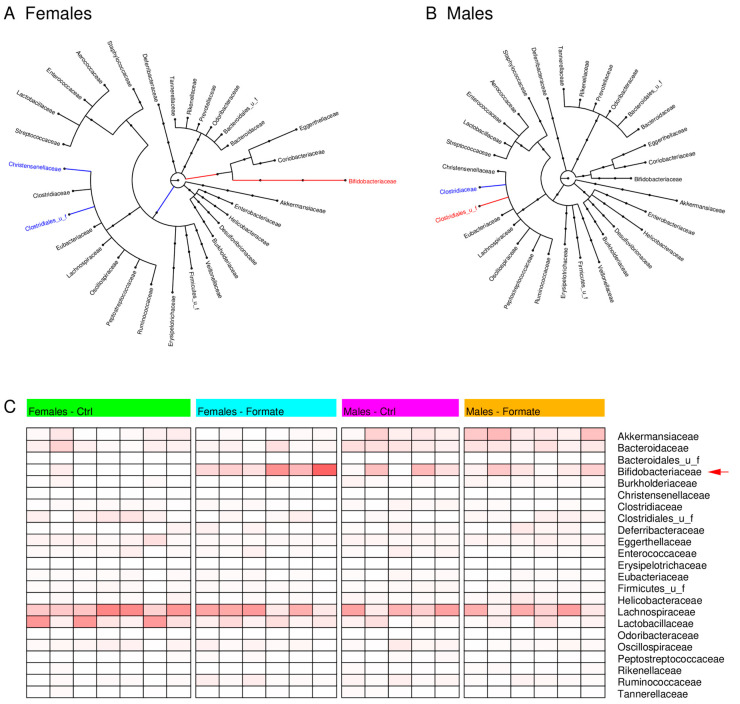
(**A,B**) Phylogenetic tree of the identified microbiota truncated at the family level, from the stools of female (**A**) and male (**B**) mice. The longer the edges the more significant is the difference of the microbe frequency between the cohort supplemented with 125 mM of sodium formate and controls. The red colour highlights a significant frequency increase by sodium formate supplementation and the blue colour a significant decrease. (**C**) Heatmap reporting the microbiota frequencies at the family level.

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
