# Peer review of "Impact of Formate Supplementation on Body Weight and Plasma Amino Acids"

_nutrients, 2020, doi:10.3390/nu12082181_

Round 1

Reviewer 1 Report

Congratulations on the very good work of the authors. The paper presents a study using modern research techniques and interestingly presents the results.
Nevertheless, the authors did not avoid a few minor omissions. First of all, the paper lacks basic data to estimate the given dose, it is necessary to include water and feed consumption by animals during the experiment. It also seems that a much more appropriate way would be to administer formic acid through a stomach cannula (animals then receive a strictly defined dose per body weight).
Please also refer in this paper to the DL50 value for the compound and the route of exposure (the authors have kept this value silent) and it seems to be similar to the applied dose.
Administration of water or feed corrosive compounds may cause histopathological changes in the upper parts of the digestive system. Was the histopathological evaluation of tissues of exposed animals carried out? If so, please include the results of this investigation.
The statistical analysis was treated very briefly. The most simple test for independent groups was used, but before its application, it is also necessary to at least check the normal distribution of the dependent variable and the homogeneity of variance. Moreover, in the case of the Student's t-test for dependent samples belongs to the family of parametric tests, so it is important that the assumption about the normality of the distribution of the dependent variable in the first and second measurements is fulfilled (which is not so obvious). Please refer to the statistics and specify which software you used. In general, I think that statistics are the weakest element of the work and should be repeated with a better selection of statistical tests.
The discussion omitted a very important thread that affects the conclusions. The authors did not refer in their conclusions to the differences in formic acid pharmacokinetics between rodents and mammals. Publications on this topic can be found without any problem, e.g. in Pubmed (Sweeting et al., 2010). I suggest that the authors refer to the results obtained in the context not only of toxicity but also of differences in kinetics which indicate that rodents such as mice, in this case, are not the best research model and the conclusion should also reflect these differences.

With best regards

Author Response

First of all, the paper lacks basic data to estimate the given dose, it is necessary to include water and feed consumption by animals during the experiment. It also seems that a much more appropriate way would be to administer formic acid through a stomach cannula (animals then receive a strictly defined dose per body weight).

Response: We went back to the experiment log files and pulled out the data for the drinking rate of mice receiving sodium formate. We found no significant differences between the water drinking rates of the female and male mice in our cohorts (new Fig. 1). Based on these data and the concentration of sodium formate in the drinking water we estimated actual sodium formate consumption rate. We thank the reviewer for the suggestion of administer formic acid through a stomach cannula. We will consider that route of administration in future studies.

Please also refer in this paper to the DL50 value for the compound and the route of exposure (the authors have kept this value silent) and it seems to be similar to the applied dose.

Response: We have reported the LD50 for sodium formate in the new manuscript. While our daily dose of sodium formate is on the range of the LD50, this dose is distributed along several small doses during the 24 hour day/night. Furthermore, no mouse died during the 4 months of sodium formate supplementation.

Administration of water or feed corrosive compounds may cause histopathological changes in the upper parts of the digestive system. Was the histopathological evaluation of tissues of exposed animals carried out? If so, please include the results of this investigation.

Response: Unfortunately, we did not plan for this assessment. We thank the reviewer for the suggestion that we will consider in future studies.

The statistical analysis was treated very briefly. The simplest test for independent groups was used, but before its application, it is also necessary to at least check the normal distribution of the dependent variable and the homogeneity of variance. Moreover, in the case of the Student's t-test for dependent samples belongs to the family of parametric tests, so it is important that the assumption about the normality of the distribution of the dependent variable in the first and second measurements is fulfilled (which is not so obvious). Please refer to the statistics and specify which software you used. In general, I think that statistics are the weakest element of the work and should be repeated with a better selection of statistical tests.

Response: We have applied the Shapiro-Wilk test and it does not reject the normal distribution for both control and cases in all instances where a significant difference was reported. The test for homogeneity of the variance is not required because we applied an unequal variance t-test, also known as Welsh test. We have also expanded the methods section describing the statistics to report more details about the tests performed and the software used. In our opinion, the applied statistical test is sufficient to support the conclusions being made.

The discussion omitted a very important thread that affects the conclusions. The authors did not refer in their conclusions to the differences in formic acid pharmacokinetics between rodents and mammals. Publications on this topic can be found without any problem, e.g. in Pubmed (Sweeting et al., 2010). I suggest that the authors refer to the results obtained in the context not only of toxicity but also of differences in kinetics which indicate that rodents such as mice, in this case, are not the best research model and the conclusion should also reflect these differences.

Response: We agree with the reviewer that the differences between humans and mice should be highlighted. We have already done so in the last paragraph of the discussion section, specifying that there are differences between mice and humans regarding the rate of formate metabolism. In particular, we emphasize that the formate demand is much higher in mice, which basically means that the pharmacokinetics is faster in mice. However, we disagree with the reviewer comment that from this evidence we can conclude that the mouse is not a good model. In our opinion the mouse is a valid model as long as we work with the right dose scaling between organisms. To our knowledge a lot of the whole-body formate metabolism we know comes from studies in mice, and that is independent of the fact that mice and human have different tolerances to methanol or formate.

Reviewer 2 Report

There are few studies on the physiological impact of one-carbon nutritional sources. It is also apparent that most, if not all, of them relate to dietary restriction but not supplementation with one-carbon sources. Despite the diversity of one-carbon nutritional sources, most studies to date have focused on how the dietary restriction of the essential amino acid methionine -the reduction of which has anti-aging and anti-obesogenic properties- influence also cancer outcomes via controlled and reproducible changes to one-carbon (1C) metabolism.

To the best of my knowledge, the work presented now by Dhayade et al. constitutes the first study assessing the physiological consequences of dietary supplementation with the one-carbon precursor formate. The authors convincingly propose that formate supplementation induces significant physiological changes in a gender-specific manner (i.e., solely in females), including:

  • Increase in body weight
  • Reduction of plasma amino acids
  • Increase of health-promoting gut microbiome (i.e., bifidobacteria).

I have a few comments that the authors should consider, mainly to increase its relevance.

It may seem paradoxical to the reader that, in terms of cancer prevention and/or treatment but likely also in the prevention/treatment of other metabolic disorders (obesity), two opposite strategies (methionine deprivation and formate supplementation) that alter the pools of one-carbon units would provide the same healthy phenotypic outcome. The authors should elaborate a model in which integrate the methionine deprivation versus formate supplementation both as health-promoting strategies in terms of 1C metabolism; particularly, based on their current findings, the authors should speculate about the mechanisms explaining the mechanistic basis of low serum formate levels in cancer patients and obese individuals and if they would expect a completely different physiological outcome of formate supplementation if, instead of employing metabolically-normal (animal) models, they would have used metabolically altered models (e.g., high-fat diets to study rodent obesity, tumor-prone transgenic models, etc).

If the supplementation of dietary formate does not drastically impact physiology of metabolically-normal models, how the authors assimilate that the occurrence of formate overflow not only in cancer cells but likely also in T-cells and other metabolically-specialized tissues/cells drives low levels of circulating formate (that, supposedly, should be corrected/returned to normality)?

Bifidobacteria are known to alter their metabolic pathways based upon the carbohydrates that are available for their use (Palframan et al. Curr Issues Intest Microbiol. 2003;4(2):71-5, i.e., high lactate production correlated with low formate production and low lactate concentrations were obtained at higher levels of formate). Have the author looked at the impact of formate supplementation on circulating glucose/lactate in these animals?

Author Response

It may seem paradoxical to the reader that, in terms of cancer prevention and/or treatment but likely also in the prevention/treatment of other metabolic disorders (obesity), two opposite strategies (methionine deprivation and formate supplementation) that alter the pools of one-carbon units would provide the same healthy phenotypic outcome. The authors should elaborate a model in which integrate the methionine deprivation versus formate supplementation both as health-promoting strategies in terms of 1C metabolism; particularly, based on their current findings, the authors should speculate about the mechanisms explaining the mechanistic basis of low serum formate levels in cancer patients and obese individuals and if they would expect a completely different physiological outcome of formate supplementation if, instead of employing metabolically-normal (animal) models, they would have used metabolically altered models (e.g., high-fat diets to study rodent obesity, tumor-prone transgenic models, etc).

Response: This is indeed an interesting point. Based on our data for female mice, formate supplementation lowers plasma amino acids, including methionine. In turn, we have evidence that a low protein diet induces an increase in the expression of serine synthesis and mitochondrial formate production in the liver. From this evidence we could speculate that amino acids deprivation and formate supplementation may trigger similar physiological effects. We have added this text in the discussion section.

If the supplementation of dietary formate does not drastically impact physiology of metabolically-normal models, how the authors assimilate that the occurrence of formate overflow not only in cancer cells but likely also in T-cells and other metabolically-specialized tissues/cells drives low levels of circulating formate (that, supposedly, should be corrected/returned to normality)?

Response: It is known that adipocytes cultured in vitro increase the release of uric acid, which should be matched by an increase of incorporation of formate into urate (although not measured). It is also well documented that urate levels are increased in the blood of obese individuals. In fact, urate is the metabolite with the best correlation with body mass index [https://pubmed.ncbi.nlm.nih.gov/30318341/]. One hypothesis is that the increase of adipocyte tissue in the context of obesity lead to an increased the incorporation of formate into urate leading to a depletion of blood serum. Thus, low formate may be a bystander rather than a driver of obesity. Why we find this very interesting it is still very speculative.

Bifidobacteria are known to alter their metabolic pathways based upon the carbohydrates that are available for their use (Palframan et al. Curr Issues Intest Microbiol. 2003;4(2):71-5, i.e., high lactate production correlated with low formate production and low lactate concentrations were obtained at higher levels of formate). Have the author looked at the impact of formate supplementation on circulating glucose/lactate in these animals?

Response: We did not observe any association between formate supplementation and plasma glucose/lactate levels. In our opinion, the increase in bifidobacterial frequency is an independent event that could be related to the balance between oxalate and formate metabolism in bifidobacterial. We have added text in the discussion section to state our hypothesis for the observed increase in bifidobacterial frequency.

Reviewer 3 Report

This paper describes the physiological effects of a several-month formate supplementation regime in male and female mice. It is known that formate metabolism is required for embryonic development but less is understood about the role of formate in adult metabolism. This paper uses a variety of techniques to assess the effect of formate supplementation on male and female mouse immune function, metabolism, body mass and the microbiome. The results of the study are largely descriptive and the authors do not provide an explanation for the observed differences. While the hesitancy to speculate is understood, an attempt to describe what could be going on or what next steps for study might be would be useful.

Comments:

Line 47: “Based on the daily urine excretion of uric acid we need about 1 gram of formate per day.” – What is this statement based on? Reference 7 is the lead author’s self-published book that is not widely available nor peer reviewed.

Line 54: The bacterial profile in the female mice may be altered by formate, but is it clear that it is a more pro-biotic bacterial profile?

Line 67: Was the pH adjusted after addition of sodium formate?

Line 72: The authors state that allantoin is assumed to be a major sink for formate in urine. What assumption based on - is allantoin as a major sink backed up by a study? If so, please provide the reference.

Line 75: What is the assumption of urinary output of 1 mL a day based on?

Lines 126-138: Has the assay described here been published previously? How was the assay validated for inter- and intra-assay variability?

Line 156 describing the gating strategy: The gating strategy appears to be well thought out but it is beyond my area of expertise to assess if the strategy is sound.

Line 336: Reference 5 is missing volume, pages

Line 339: Reference 7 is a self-published book and is not widely available. A more appropriate reference should be used.

Figure 1: How long was the treatment? The Methods section states "approximately 4 months". Please state in legend.

There is no difference in the immune cell markers investigated so apparently the increased spleen size is not associated with an increase in immune activity. Is there any explanation for the difference in spleen size between formate treated and control females? Similarly, what could be the explanation for the difference in gut bacteria and plasma proteinogenic amino acids in the formate-treated female mice. Do bifidobacteria preferentially consume formate over other types of gut-dwelling bacteria, or is there some other potential metabolic connection?

Author Response

The results of the study are largely descriptive and the authors do not provide an explanation for the observed differences. While the hesitancy to speculate is understood, an attempt to describe what could be going on or what next steps for study might be would be useful.

Response: We have added new text in the discussion section with hypotheses that explain our data.

Line 47: “Based on the daily urine excretion of uric acid we need about 1 gram of formate per day.” – What is this statement based on? Reference 7 is the lead author’s self-published book that is not widely available nor peer reviewed.

Response: As stated, it is based on the daily urine excretion of uric acid. There is no other reference making an estimate of the human daily demand of formate. Therefore, there is no other citation possible. We also note that this value is only brought as a reference for the discussion. It does not affect the analysis of the reported data.

Line 54: The bacterial profile in the female mice may be altered by formate, but is it clear that it is a more pro-biotic bacterial profile?

Response: We have made this statement more precise by reporting that formate supplementation increases the frequency of bifidobacterial, a probiotic bacteria.

Line 67: Was the pH adjusted after addition of sodium formate?

Response: No we did not attempted to adjust the pH. The pH of sodium formate solutions (up to 1M) is similar to the water pH, as indicated by the sodium formate provider https://www.sigmaaldrich.com/catalog/product/sigma/71539?lang=en&region=GB

Line 72: The authors state that allantoin is assumed to be a major sink for formate in urine. What assumption based on - is allantoin as a major sink backed up by a study? If so, please provide the reference.

Response: Mice as a difference with humans metabolise urate to allantoin, and the latter is found at higher con concentrations in the urine. We have added a reference.

Line 75: What is the assumption of urinary output of 1 mL a day based on?

Response: This assumption is based in general knowledge about C57BL mice. We have added a citation as a reference. We note this value was used during the design of the study, but it is not needed during the analysis of the data.

Lines 126-138: Has the assay described here been published previously? How was the assay validated for inter- and intra-assay variability?

Response: We have been using this assay although we have not reported a study about the assay validation. Based on experience most of the variability we observe comes from physiological variations across mice with the same genetic background. The variations due to the assay are much smaller.

Line 156 describing the gating strategy: The gating strategy appears to be well thought out but it is beyond my area of expertise to assess if the strategy is sound.

Response: Two of the authors (RW and SC) are experts on this experimental technique and they acknowledge that it is correct.

Line 336: Reference 5 is missing volume, pages

Response: Corrected.

Line 339: Reference 7 is a self-published book and is not widely available. A more appropriate reference should be used.

Response: There is no more appropriate reference, since there is no other reference making an estimate of daily needs of formate.

Figure 1: How long was the treatment? The Methods section states "approximately 4 months". Please state in legend.

Response: We have made the statement in the methods more precise: “for 127 days, approximately 4 months”. We have also added these numbers in the Figure 1 legend to make it more accessible to the reader.

There is no difference in the immune cell markers investigated so apparently the increased spleen size is not associated with an increase in immune activity. Is there any explanation for the difference in spleen size between formate treated and control females? Similarly, what could be the explanation for the difference in gut bacteria and plasma proteinogenic amino acids in the formate-treated female mice. Do bifidobacteria preferentially consume formate over other types of gut-dwelling bacteria, or is there some other potential metabolic connection?

Response: We went back to the data and calculated the spleen weight relative to body weight. The data is shown in the new Figure 1E. Relative to body weight, there are no significant changes in the spleen weight. That is although the spleen grows heavier in the female mice supplemented, this weight increase is in proportion to the increase in body weight. This evidence suggests that there is an overall increase of body weight and organ weights, but all in proportion. It is unfortunate that we did not weight other organs to further support this hypothesis and we would consider that in future studies.

Round 2

Reviewer 1 Report

Dear Authors

In my opinion, you have not applied correct statistical tests. In case of checking statistical differences between several groups the t-student test cannot be used, in such cases you should be use e.g. ANOVA type statistics.

Author Response

In my opinion, you have not applied correct statistical tests. In case of checking statistical differences between several groups the t-student test cannot be used, in such cases you should be use e.g. ANOVA type statistics

Response: We understand that it is more common to carry on a multi-group ANOVA when there are two or more variables (here gender and formate supplementation). However, we are not checking statistical differences between several groups because we are considering the males and females cohorts as two independent experiments. The reason to do so is that the differences in the response to formate supplementation are quite extreme between the female and male mice. In our opinion, in the context of our experiments, the females and male cohorts should be kept apart and should not be analysed together using the linear models approach of multi-group ANOVA. It would be like mixing data from mice and rats cohorts.